# Factors associated with resistance to SARS-CoV-2 infection discovered using large-scale medical record data and machine learning

Kai-Wen K. Yang[1,2☯], Chloé F. Paris[1,2☯], Kevin T. Gorman[1,2☯], Ilia Rattsev[1,2☯], Rebecca H. Yoo[1,2☯], Yijia Chen[1☯], Jacob M. Desman[1,2], Tony Y. Wei[1,2], Joseph L. Greenstein[1,2], Casey Overby Taylor[1,2,3], Stuart C. Ray[4]*

**1** Department of Biomedical Engineering, Johns Hopkins University, Baltimore, MD, United States of America, **2** Institute for Computational Medicine, Johns Hopkins University, Baltimore, MD, United States of America, **3** Department of Medicine, Division of General Internal Medicine, Johns Hopkins School of Medicine, Baltimore, MD, United States of America, **4** Department of Medicine, Division of Infectious Diseases, Johns Hopkins School of Medicine, Baltimore, MD, United States of America

☯ These authors contributed equally to this work.
* sray@jhmi.edu

**Data Availability Statement:** Our dataset cannot be shared publicly due to IRB restrictions on data obtained from participants without consent for

## Abstract

There have been over 621 million cases of COVID-19 worldwide with over 6.5 million deaths. Despite the high secondary attack rate of COVID-19 in shared households, some exposed individuals do not contract the virus. In addition, little is known about whether the occurrence of COVID-19 resistance differs among people by health characteristics as stored in the electronic health records (EHR). In this retrospective analysis, we develop a statistical model to predict COVID-19 resistance in 8,536 individuals with prior COVID-19 exposure using demographics, diagnostic codes, outpatient medication orders, and count of Elixhauser comorbidities in EHR data from the COVID-19 Precision Medicine Platform Registry. Cluster analyses identified 5 patterns of diagnostic codes that distinguished resistant from non-resistant patients in our study population. In addition, our models showed modest performance in predicting COVID-19 resistance (best performing model AUROC = 0.61). Monte Carlo simulations conducted indicated that the AUROC results are statistically significant ($p < 0.001$) for the testing set. We hope to validate the features found to be associated with resistance/non-resistance through more advanced association studies.

## Introduction

COVID-19 is a highly transmissible infection caused by the severe acute respiratory syndrome coronavirus 2 (SARS-CoV-2), with 621 million cases worldwide as of October 17, 2022, resulting in 6.5 million deaths [1]. SARS-CoV-2 can be transmitted between humans via respiratory droplets and close contact. All populations are generally susceptible to SARS-CoV-2. Prior studies indicate that the mean secondary attack rates for SARS-CoV-2 infections can range from 16% to 32%, which is higher than those of SARS and MERS [2–4]. Despite the high spread of infection amongst shared households, there are still some highly exposed individuals

sharing publicly. Researchers who wish to collaborate in analysis of JH-Crown data will need to collaborate with a Johns Hopkins University investigator and obtain IRB approval. Those who are requesting access to the JH-CROWN data should contact the PI, Dr. Garibaldi, at bgariba1@jhmi.edu.

**Funding:** The data utilized were part of JH-CROWN: The COVID PMAP Registry, which is based on the contribution of many patients and clinicians and is funded by Hopkins inHealth, the Johns Hopkins Precision Medicine Program. Project-specific costs of data extraction were defrayed by funds from the Office of the Dean, JHU School of Medicine. The funders had no role in study design, data collection and analysis, decision to publish, or preparation of the manuscript.

**Competing interests:** The authors have declared that no competing interests exist.

who do not contract the virus, indicating possible resistance to infection. This suggests that natural resistance against SARS-CoV-2 is present in human populations. The difference in mortality between different geographical locations also indicates that in addition to societal/behavioral factors, genetic and environmental differences may influence host defense against SARS-CoV-2 infection [5]. A recent study of *All of Us* participants showed that built environment of houses with a shared component increased the rate of SARS-CoV-2 transmission [6]. However, the elderly and individuals with underlying diseases or low immune function are more likely to develop severe COVID-19 [7]. Epidemiological studies have shown that elderly patients are more susceptible to severe disease, while children tend to have milder symptoms [8]. Host response to SARS-CoV-2 ranges from minimal symptoms to severe respiratory failure with multiple organ failure. Seibold et al. recently found that asthma is not associated with household SARS-CoV-2 transmission, but food allergy is associated with lower household SARS-CoV-2 transmission [9].

Knowing the factors that contribute to an individual's resistance to COVID-19 may facilitate our understanding of the mechanisms of viral infection and disease progression. The presence or absence of immune response phenotypes among COVID-19 resistant people may also provide clues to the pathogenesis of COVID-19. In addition, identifying mechanisms of resistance may help the research community to identify potential therapeutic targets to treat COVID-19. When the factors responsible for resistance to human immunodeficiency virus (HIV) were identified in the early 2000s, a better understanding of the HIV infection process was achieved and it opened many possibilities in the pharmacological field, such as the development of treatments other than anti-viral drugs [10]. We believe that a better understanding of SARS-CoV-2 infection may lead to similar developments in anti-SARS-CoV-2 drug discovery.

Furthermore, given the high burden of the COVID-19 pandemic on the healthcare system, identifying individuals resistant to infection may improve the quality of surveillance and help allocate healthcare resources more efficiently. It is also important to distinguish between individuals resistant to infection and those who get infected but remain asymptomatic. Resistant individuals may be less likely to transmit the virus than asymptomatic infected individuals, although further studies are needed to validate this statement. Therefore, it is important to be able to classify individuals as COVID-19-resistant or non-resistant with high predictive power.

Although there have been many studies assessing risk factors for various levels of severity of COVID-19, to our knowledge, there have been few published reports on resistance to SARS-CoV-2 infection [11–15]. A recent study has found that COVID-19 severity is associated with decreased expression of OAS1 gene, which is a part of the innate immune response to viral infections [16]. However, this study only included hospitalized and non-hospitalized patients with laboratory-confirmed SARS-CoV-2 infection of European and African ancestries [16]. Several independent studies have determined that individuals with blood type O may be less susceptible to SARS-CoV-2 infection [11–13]. Another group has examined the relationship between HLA haplotypes and susceptibility to COVID-19. HLA-B*15:03 has been found to be protective against the infection and can potentially indicate a resistance marker [14]. However, these studies did not assess the exposure to the virus among the tested individuals and performed their analyses comparing the rates of positive test results among different blood types or HLA haplotype cohorts. A recent multicenter study assessed the susceptibility to SARS-CoV-2 infection based on polymorphisms in the ACE2 receptor, a well-known 'port of entry' for the virus into human cells [15]. The major limitation of this study, however, was the lack of data on clinical outcomes at the population scale [15]. Yu et al. assessed immune memory to common cold coronaviruses (CCCs) before the COVID-19 pandemic. They suggested that repeated exposure to or infections from these CCCs might influence COVID-19 severity and

high CCC T cell reactivity is associated with pre-existing SARS-CoV-2 immunity [17]. However, the previous CCC exposure history of the cohort has limited the findings.

To address the gaps mentioned therein, our study aims to (1) explore a broader range of phenotypes that may be associated with COVID-19 resistance on a large scale (2) construct and train machine learning models to predict COVID-19 resistance. To the authors' knowledge, this will be the first study to consider exposure while defining the cases of resistance and predict COVID-19 resistance using machine learning approaches.

## Materials and methods

The overall workflow of this study is depicted in Fig 1. Our study began with data extraction, and was subsequently followed by cohort selection and data processing, clustering, and predictive modeling. The data was first extracted from the Johns Hopkins COVID-19 Precision Medicine Analytics Platform Registry (JH-CROWN). After data extraction, cohort selection and data processing were conducted, which involved defining the exposure and resistance measures, labelling the patients with these measures, and subsequent data cleaning. Then, we implemented clustering methods, which includes dimensionality reduction, pattern-based clustering, and enrichment analysis. Finally, we conducted predictive modeling, where we selected features, trained and evaluated the models.

### Cohort selection

In order to evaluate the factors associated with resistance and non-resistance to COVID-19 in patients, we conducted a retrospective analysis using Epic patient records from the Johns Hopkins COVID-19 Precision Medicine Analytics Platform Registry (JH-CROWN) [18]. The

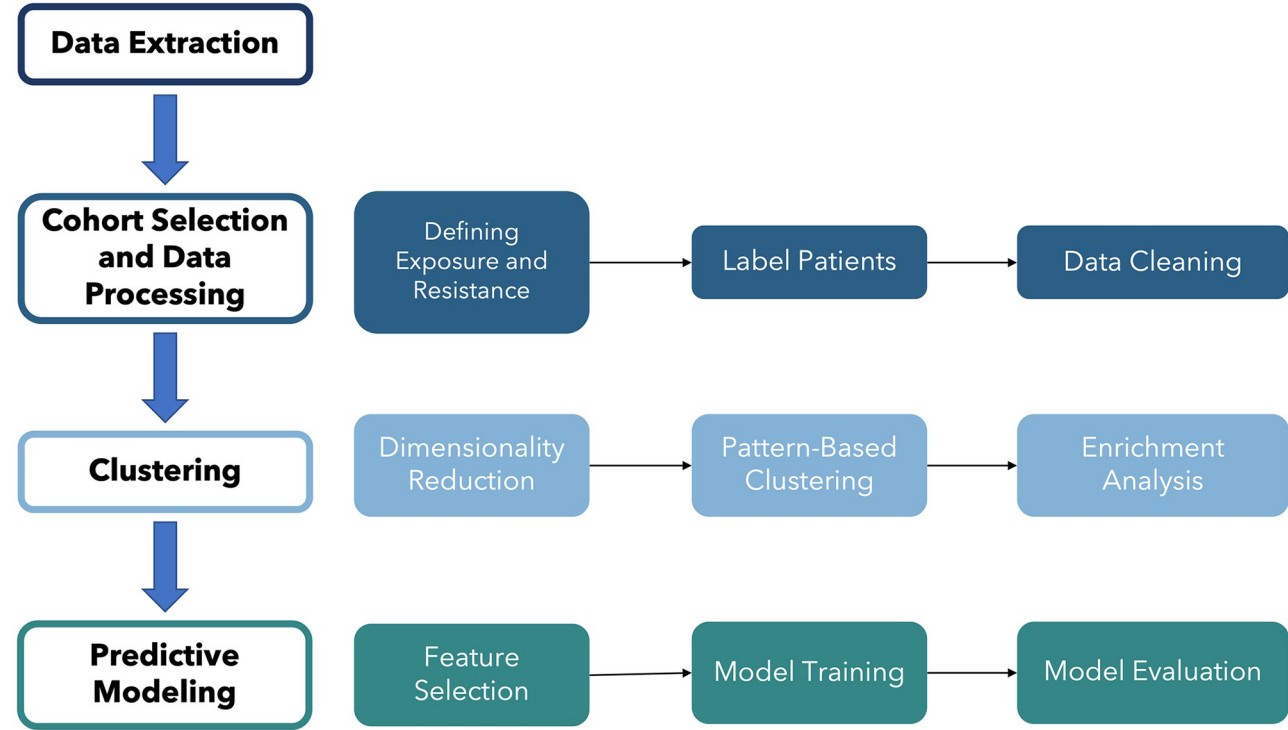

**Fig 1. Workflow of the overall study.** Workflow graph detailing the overall process with details for each step. The workflow includes data extraction, cohort selection and data processing, clustering, and predictive modeling.

JH-CROWN registry is a collection of information about patients who have been seen at a Johns Hopkins Medical Institution facility and who are suspected or confirmed of having a COVID-19 infection [19]. The de-identified EHR data in the JH-CROWN registry was accessed through the Precision Medicine Analytics Platform (PMAP) [20]. Our study protocol and registry access were approved by the Johns Hopkins Medicine Institutional Review Board (IRB00308622).

Out of the participants in the registry, only individuals who received a COVID-19 test between June 10, 2020 and December 15, 2020 were included. June 10, 2020 was chosen as a cutoff as this was the first date when the EHR prompt "reasons for testing" was recorded in JH-CROWN as part of the COVID-19 testing participant survey. The second cutoff date of December 15, 2020 was chosen to avoid the confounding effects of COVID-19 vaccines, as large-scale vaccination efforts were initiated on December 14, 2020 [21]. To identify patients that had previous exposure to SARS-CoV-2, only individuals who had at least one COVID-19 test with "potential exposure to the virus" as the reason for testing were included.

The study cohort with self-reported exposures was then divided into participants with low and high confidence exposures using household index (HHI). The household index is an identifier assigned to each person based on their geolocation (longitude and latitude) in their medical record. Individuals who share the same HHI are considered to be within the same household. The low confidence exposure cohort consists of participants who do not share a household with any individuals with a positive COVID-19 test or are in a household with 10 or more participants. We chose the threshold of 10 as a household of 10 or more is likely to be a multi-unit complex, such as an apartment or dormitory, which does not necessarily lead to a high degree of exposure. The high confidence cohort consists of participants who share a household with 10 or fewer individuals and at least one individual in the household has a positive COVID-19 test. The low-confidence exposure cohort was used as the training and testing set, while the high confidence exposure cohort was used as an additional testing test, which we title the HHI testing set. These cohorts were then further stratified into resistant and non-resistant cohorts, where resistant participants were defined as those who had only negative COVID-19 tests, and non-resistant participants defined as those who had at least one or more positive COVID-19 tests.

## Data processing

The patient demographic characteristics included in the analysis were age (as of March 1, 2020), sex, and race, as recorded in the medical records. Clinical characteristics included International Classification of Diseases 10th Revision (ICD-10) diagnostic codes and outpatient medication orders, as recorded in the EHR. We note that we chose to use ICD-10 codes, which are curated and used for billing, as we believe they help making our data more reliable. To only include the conditions the patient had at the time of exposure, we limited the set of ICD-10 codes to the ones that were recorded in PMAP for encounters between January 1, 2020 and December 31, 2020. In addition, we excluded the codes containing either the word "covid" or "coronavirus", as well as the following COVID-19-related ICD-10 codes: Z20.82, Z11.52, Z86.16, J12.82 and U07.1. The set of medications included for the analysis was reduced to outpatient medications taken within 20 days before or after testing for suspected exposure to account for incubation and infectious periods.

## Pattern selection and clustering

To identify patterns and cluster our resistant cohort, we utilized the Maximal-frequent All-confident pattern Selection (MAS) Pattern-based Clustering (PC) algorithm [22]. MASPC has

been specifically designed for EHR data analysis that combines demographic information with diagnostic codes and has been described in detail by Zhong et al [22]. In short, the algorithm consists of two main phases: pattern selection (MAS) and subsequent clustering (PC). In the MAS phase, the algorithm takes a set of patient diagnostic codes and selects the subsets of diagnostic codes (named patterns) based on three user-specified parameters: the minimum proportion of records diagnostic codes need to occur in (*minSup*), minimum confidence of correlation between diagnostic codes (*minAc*), and when the patterns share diagnostic codes, they need to co-occur in at least (*minOv*) records. After the patterns of diagnostic codes are identified, the algorithm constructs a binary representation of the dataset through the use of one-hot encoding, combines it with binary represented demographic data, and applies agglomerative average-linkage hierarchical clustering based on cosine similarity [22].

For the purposes of our study, we combined ICD-10 diagnostic codes and medications into one set of clinical features and applied the MASPC algorithm. We selected *minSup* and *minAc* values that maximized the coverage of participants having at least one pattern. Based on our exploration, the optimal values for *minSup* and *minAc* were 0.08 and 0.08, respectively, meaning that any clinical feature considered for inclusion into a pattern was observed in at least 8% of records, and at least 8% of records have all features included in the pattern. We kept the *minOv* parameter at the default value of 3, as changing it did not affect coverage. We set the number of clusters *k* equal to 3, as increasing the number of clusters only increased the separation based on demographic features rather than patterns.

We then compared the prevalence of identified patterns in resistant and non-resistant individuals in an enrichment analysis. Fisher's exact test was used to estimate significance of the differences between cohorts ($\alpha = 0.05$).

## Predictive model building

We hypothesized that the resistance to COVID-19 can be predicted based on the presence or absence of specific phenotypes, and therefore, aimed to build machine learning models to predict resistance in individuals who have been exposed to SARS-CoV-2. Input features to the model include patient demographics (age, sex, race), ICD-10 diagnostic codes, outpatient medication orders, and the count of Elixhauser Comorbidities. ICD-10 codes were trimmed down to retain only the first three characters that designate the category of the diagnosis. The pharmaceutical class which indicates the chemical families the drug belongs to was extracted for each medication a patient received. Subjects without Elixhauser Comorbidity Index records were imputed with the median value of the training dataset. Categorical variables were one-hot encoded to represent their presence in a subject's record, while discrete variables (age and count of Elixhauser Comorbidities) were normalized to have values between 0 and 1.

The low-confidence dataset was spilt into training and testing subsets (8:2). To decrease sparsity, any feature with less than five occurrences in the training dataset was removed. To reduce feature dimension, recursive feature elimination with L2 regularization was utilized to select the top 108 features (out of 1,310 features) to be included in model training. Given the high imbalance between the two classes (resistant vs. non-resistant), various re-sampling techniques were explored, including adjusting the weights of the model, under-sampling the majority class, and up-sampling the minority class through the Synthetic Minority Over-sampling Technique (SMOTE) [23].

Several commonly used models, including logistic regression and random forest from sklearn [24] and XGBoost [25] were trained with the training dataset, with model hyperparameters tuned using a grid search with five-fold cross-validation [26]. The optimized models were evaluated both on the testing subset from the low-confidence exposure cohort and the

HHI testing subset. Monte Carlo simulations were conducted to evaluate the significance of the model results. In addition, SHapley Additive exPlanations (SHAP), a method that uses optimal credit allocations among entities to derive their contributions [27], was employed to evaluate the relative importance of each feature, enabling us to understand how feature values affect the prediction outcomes across the entire population. We choose the best-performing model based on which one performed the most consistently well on the held-out data across AUROC, accuracy, sensitivity, and specificity to be used for this SHAP analysis.

The HHI testing subset was held out of all stages of experiment design, model tuning, and model training in order to reduce observer bias. Decisions about patient inclusion were made prior to receiving the data to minimize exclusion bias. Feature selection was done through backward feature elimination on the basis of model accuracy improvement to keep researchers blind to predictors until they were examined at the end of model training.

## Results

### Cohort summary statistics

The cohort selection flowchart and the inclusion/exclusion criteria can be seen in Fig 2. We assumed that the absence of a given diagnostic code or a medication in a record was indicative of a patient not having that ICD-10 code or medication, rather than of missing data. The only field where we observed missing values was Elixhauser Comorbidity data, with 19 patients (<1%) missing.

Using the final dataset from our inclusion/exclusion criteria, we compared the participant characteristics between the low and high-confidence cohorts, further stratified by resistance and non-resistance. The descriptive statistics for demographic and key clinical characteristics are presented in Table 1. We observed similar demographic distributions of proportions for sex, age group, and race between the resistant and non-resistant cohorts within each exposure group; however, there were slightly different distributions between the exposure groups, especially for race. In the low-confidence group, the majority of participants were White, while in the high-confidence group, there was an approximately equal proportion of White and Black

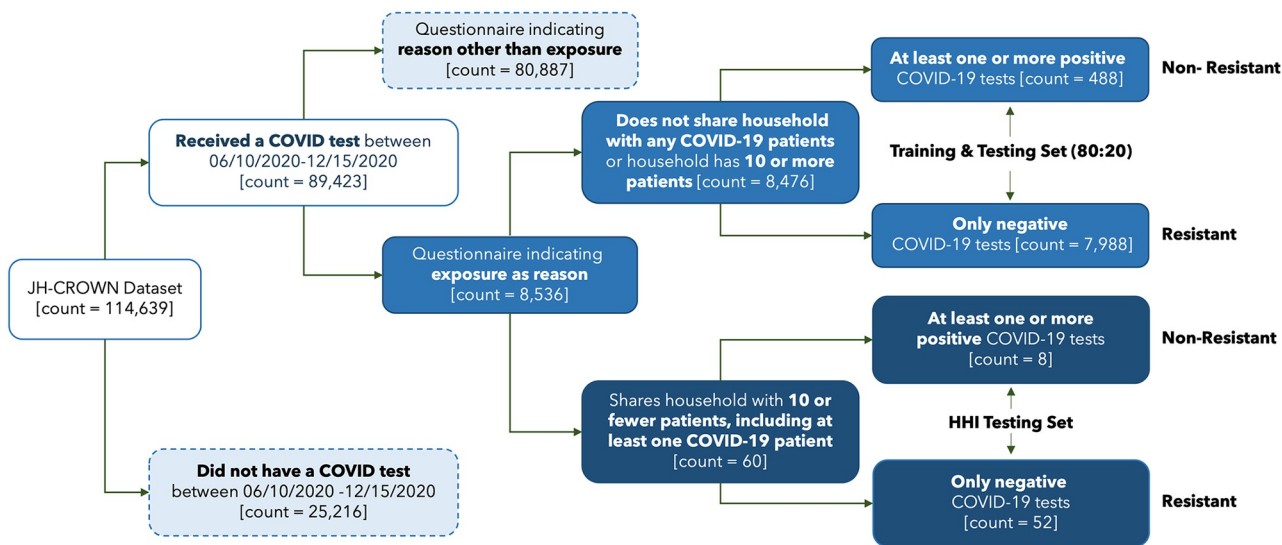

**Fig 2. Cohort selection flowchart.** Participants from the JH-CROWN are stratified into different cohorts, with four final cohorts: a low confidence of exposure cohort (resistant/non-resistant) and a high confidence of exposure cohort (resistant/non-resistant).

**Table 1. Summary statistics for the low-confidence and high-confidence exposure groups broken down by resistance.** Number of patients (percent of total population) is reported for categorical variables; mean (standard deviation) is presented for continuous variables.

| | | Low-Confidence Exposure | | High-Confidence Exposure | |
| --- | --- | --- | --- | --- | --- |
| | | Resistant | Non-Resistant | Resistant | Non-Resistant |
| Sex | Male | 3471 (43%) | 195 (40%) | 19 (37%) | 2 (25%) |
| | Female | 4515 (57%) | 293 (60%) | 33 (63%) | 6 (75%) |
| Age Group | < 20 | 2411 (30%) | 148 (30%) | 19 (36%) | 3 (37%) |
| | 20–39 | 2183 (27%) | 126 (25%) | 17 (33%) | 2 (25%) |
| | 40–59 | 2166 (27%) | 121 (25%) | 12 (23%) | 2 (25%) |
| | 60 | 1228 (16%) | 93 (20%) | 4 (8%) | 1 (13%) |
| Race | White | 4466 (56%) | 257 (53%) | 20 (38%) | 2 (25%) |
| | Black | 2058 (26%) | 130 (27%) | 19 (37%) | 4 (50%) |
| | Asian | 482 (6%) | 26 (5%) | 4 (8%) | 0 (0%) |
| | Other | 922 (12%) | 70 (15%) | 9 (17%) | 2 (25%) |
| Unique Medications | | 2.2 (4.1) | 2.1 (3.5) | 2.9 (5.3) | 1.0 (1.4) |
| Unique Diagnoses | | 14 (16) | 16 (18) | 17 (16) | 14 (7) |
| Elixhauser Comorbidities | | 3.0 (4.5) | 3.4 (3.9) | 3.9 (4.2) | 1.7 (1.6) |

participants. The average numbers of unique medications, unique diagnoses, and Elixhauser Comorbidities per patient were reported for each group along with standard deviation. Although the low-confidence exposure group showed similar averages for unique medications, diagnoses and count of Elixhauser Comorbidities between resistant and non-resistant subgroups, there was noticeably large variability in these groups, as seen in the large standard deviations. The high-confidence exposure group also had large variations in unique medications, diagnoses and Elixhauser Comorbidities, but the resistant subgroup had more medications, diagnoses, and comorbidities on average than the non-resistant subgroup. Overall, there were similar averages across the resistant and non-resistant groups of the low-confidence exposure group and the resistant group of the high-confidence exposure group for all health measures.

## Clustering

**MASPC pattern selection.** Through MASPC on the low-confidence cohort, we identified 56 patterns, resulting in a cohort coverage of 76%. We split these 56 patterns into two groups: patterns associated with resistant individuals and patterns associated with non-resistant individuals. We then conducted a Fisher's exact test to explore pattern discrepancies between the two groups. Table 2 shows the results from this statistical test. We observed that five of our patterns had a p-value below 0.05. In addition, the clustering results from MASPC indicated that only one pattern (personal history of malignant neoplasm) was associated with SARS-CoV-2 resistance while four patterns (long term (current) drug therapy; screening for infectious and parasitic diseases, screening for infectious and parasitic diseases; encounter for immunization, disorders of fluid, electrolyte and acid-base balance, dorsalgia, and personal history of malignant neoplasm) were associated with SARS-CoV-2 non-resistance.

**MASPC clustering.** MASPC clustering was performed on the resistant cohort. It was found that clusters formed by demographic features, with the first cluster including mainly female participants, the second cluster including mainly males, and the third cluster comprising mainly of children aged 0–10, with both females and males in the cluster. Each cluster was analyzed, exploring the prevalence of the 56 patterns within the three cluster. Fig 3 visualizes some results from this exploration. We found that nicotine dependence (F17) was enriched in

**Table 2. Prevalence of patterns found using MASPC method in both resistant and non-resistant patients.** Five diagnostic code patterns were found with a p-value less than 0.05. Odds Ratios less than 1 indicate prevalence in non-resistant cohort, whereas odds ratios greater than 1 indicate prevalence in resistant cohort.

| MASPC Pattern | Odds Ratio | P-Value |
|---|---|---|
| Long-term (current) drug therapy; Screening for infectious and parasitic diseases | 0.59 | 0.0055 |
| Screening for infectious and parasitic diseases; Encounter for immunization | 0.67 | 0.01 |
| Disorders of fluid, electrolyte and acid- base balance | 0.70 | 0.021 |
| Dorsalgia (back pain) | 0.73 | 0.043 |
| Personal history of malignant neoplasm | 1.54 | 0.036 |

the cluster consisting mainly of resistant males, while depressive episodes (F32) were enriched in the cluster consisting mainly of resistant females. The pattern of long term (current) drug therapy alongside type 2 diabetes mellitus (Z79, E11) was not observed among participants belonging to the cluster comprised of children aged 0–10. However, both screening for malignant neoplasms (Z12) and asthma (J45) were prevalent in the resistant children cluster.

## Predictive modeling

The five-fold cross validation hyperparameter tuning yielded the following models. For XGBoost, learning rates from 0.01 to 0.5, between 5 and 100 estimators, and the boosters gbtree, gblinear, and dart were performed [25]. The learning rate of 0.5, 10 estimators, and the booster gbtree were selected for producing the highest cross-validation AUROC. For logistic regression, the solvers Liblinear, Newton Conjugate Gradient, and LBFGS were tried [24]. The regularizers L1, L2, and elastic net were tried, and inverse regularization strengths from $e^{-5}$ to $e^2$ were tried. The Liblinear solver, L2 penalty, and $e^{1.59}$ inverse regularization strength were selected for producing the highest cross-validation AUROC. For random forest, we examined max depths from 10 to 100, min leaf samples from 1 to 4, min samples for a split from 2 to 10, and class weights that were balanced, balanced subsample, or none. A max depth of 15, 1 min

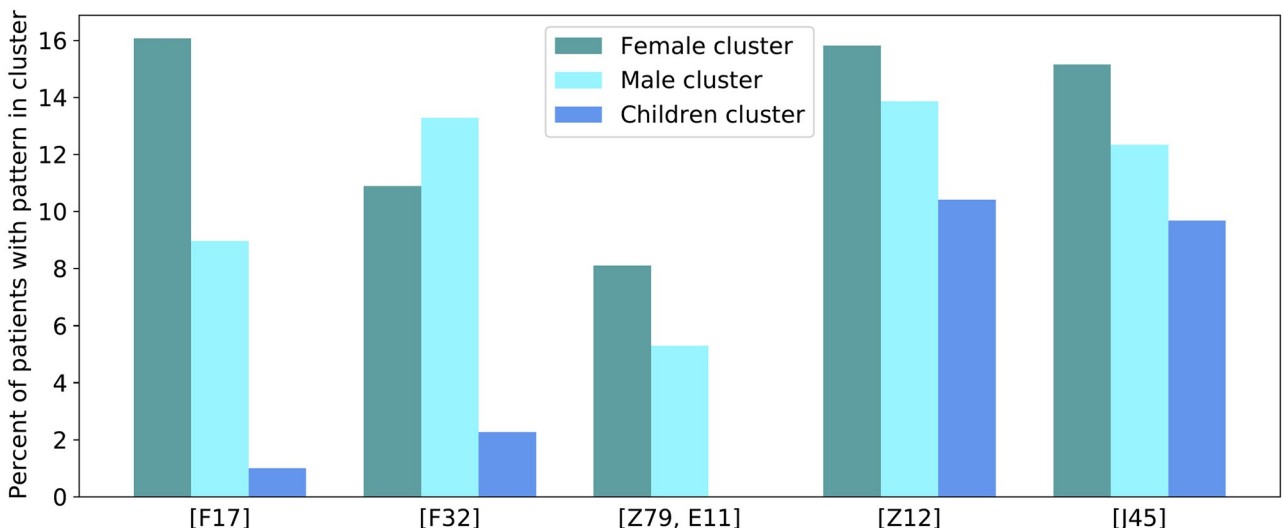

**Fig 3. Clustering results from MASPC method.** Clusters form by demographic features, with a majority of females, males, and children respectively in each of the three clusters. Shown are the distribution of patients with patterns of ICD10 codes: nicotine dependence [F17], depressive episode [F32], long term drug therapy & Type 2 diabetes [Z79, E11], screening for malignant neoplasms [Z12], and asthma [J45].

**Table 3. Predictive model performance.** XGBoost had the best model performance for both the testing set of the low-confidence group and the HHI testing set, with AUROCs of 0.61 and 0.62, respectively.

| | XGBoost | | Logistic Regression | | Random Forest | |
|---|---|---|---|---|---|---|
| | Testing | HHI Testing | Testing | HHI Testing | Testing | HHI Testing |
| **AUROC** | 0.61 | 0.62 | 0.57 | 0.50 | 0.60 | 0.53 |
| **Accuracy** | 0.63 | 0.77 | 0.77 | 0.77 | 0.49 | 0.78 |
| **Sensitivity** | 0.64 | 0.79 | 0.80 | 0.85 | 0.48 | 0.85 |
| **Specificity** | 0.57 | 0.63 | 0.32 | 0.25 | 0.69 | 0.38 |

leaf sample, 4 samples in a split, and no class weights were selected for producing the highest cross-validation AUROC.

After training the models and using five-fold cross validation to help tune model hyper-parameters, we analyzed the results of model performance on the testing set, which can be seen in Table 3 and Fig 4. XGBoost had the best model performance for both the testing set of the low-confidence group and the HHI testing set, with an AUROC of 0.61 for the testing set and 0.62 for the HHI testing set. The Monte Carlo simulations found a p value of < 0.001 when seeing how frequently an AUROC of 0.61 was achieved on a random feature using the testing set. The HHI testing set had a p value of 0.12, despite having a higher AUROC, due to its smaller population size. The logistic regression model had higher accuracy and sensitivity than XGBoost, but its much lower AUROC and specificity indicated that it may be biased in favor of making positive predictions. The XGBoost model performed consistently well across all metrics, leading us to select it for further analysis.

Shapley feature importance was performed on the testing set for the XGBoost model (Fig 5). The features that influenced the model prediction the most were ICD10 codes Z00 (encounter for general examination without complaint, suspected or reported diagnosis), Z85

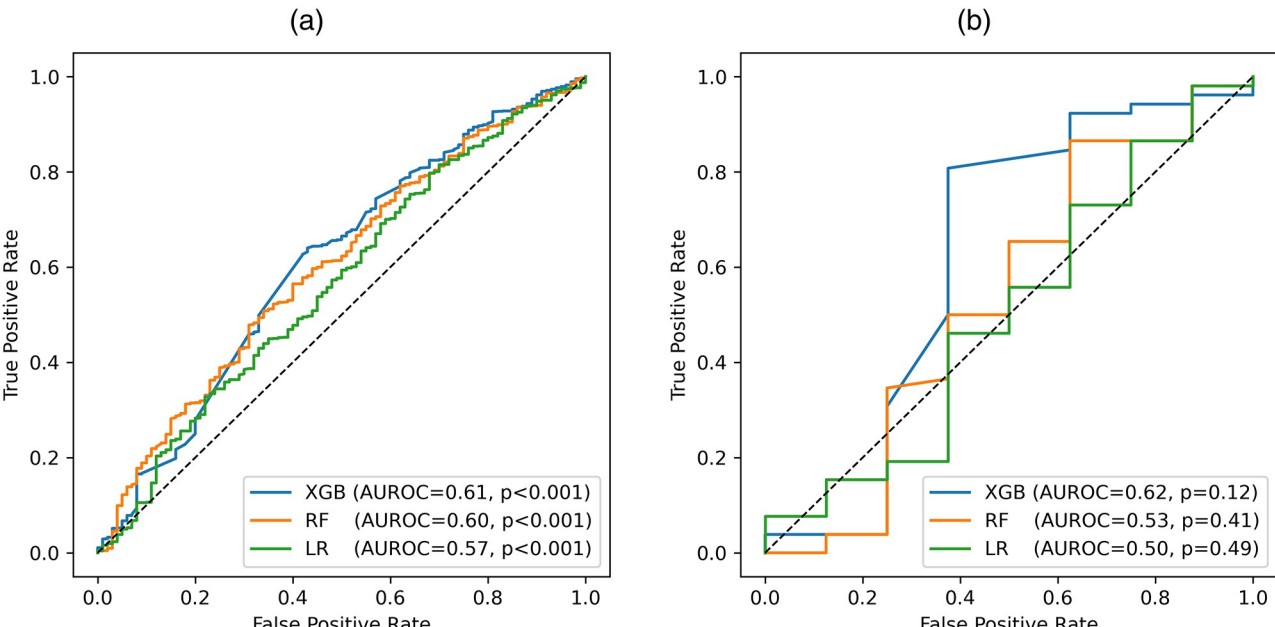

**Fig 4. Receiver operating curves of XGBoost (XGB), Random Forest (RF), and Logistic Regression (LR) models.** (a) Testing Set: XGB is the best performing model and all three models have statistically significant AUROCs (p<0.001) (b) Household Index Testing Set: XGB is again the best performing model, yet the p-values are less statistically significant due to the small sample size.

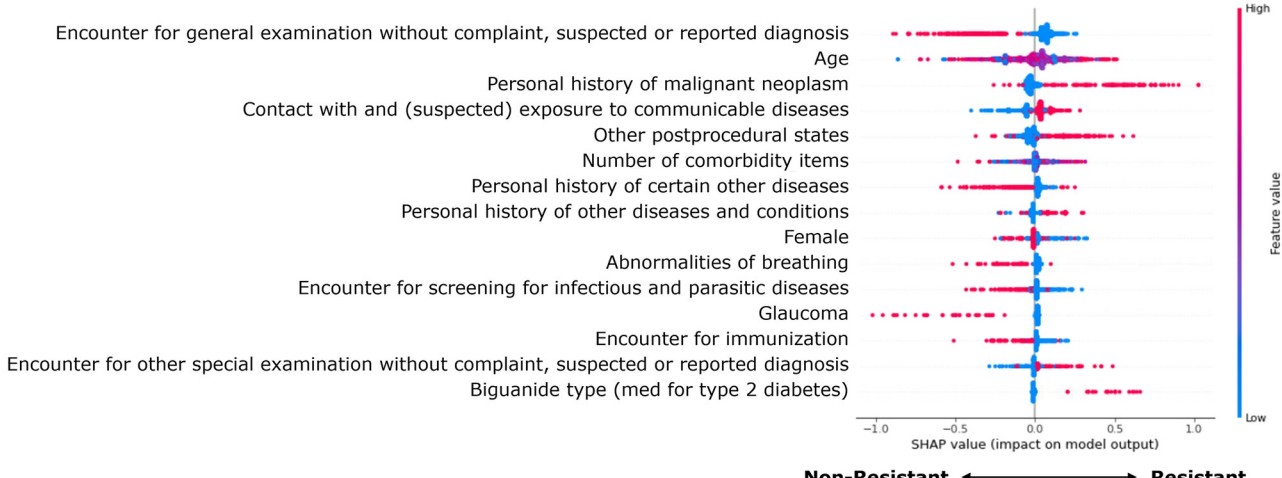

**Fig 5. Shapley feature importance of the XGBoost model.** Points on the right with positive SHAP values indicate that inclusion of the feature moves the prediction toward resistance. The color red represents a high value for the feature whereas blue indicates a low value. Features are sorted vertically by their mean absolute influence on the prediction. Bolded features are features that were also identified as important using the MASPC patterns as shown in Table 2.

(personal history of malignant neoplasm), Z20 (contact with and (suspected) exposure to communicable diseases), and Z98 (other postprocedural states), as well as age. The count of Elixhauser Comorbidities an individual had was also identified as one of the most important features by the XGBoost model.

## Discussion

In this retrospective cohort study of individuals indicating previous exposure to SARS-CoV-2, we determined risk factors associated with resistance SARS-CoV-2 infection and built a statistical model aiming at predicting individual's resistance status based on their clinical and demographic profile. Although several studies have identified possible markers of COVID-19 resistance [11–15], the major drawback of those studies was inability to distinguish between those individuals who had been exposed to the virus and those who had not. To our knowledge, this is the first study assessing COVID-19 resistance in the exposed individuals.

We examined feature correlation to avoid missing important features due to variables contributing the same information, but we didn't want to remove features that improved model performance. Our recursive feature elimination removes features that provided redundant information to the model, which includes some correlated features. We also use the broadest medication and diagnostic classes to group similar codes.

MASPC analysis revealed several patterns of diagnostic codes associated with SARS-CoV-2 resistance status. Two patterns associated with COVID-19 non-resistance contained the ICD-10 code indicating screening for infectious and parasitic disease, which was to be expected. Other signals from MASPC analysis that reduced the odds of being resistant included the diagnoses of fluid, electrolyte and acid-base balance and dorsalgia. While alkalosis has been previously reported to be common among patients diagnosed with COVID-19 [28], we were surprised to see dorsalgia in the list of emerged signals. It is of interest to validate this finding and provide a mechanistic support for the association in future research.

Surprisingly, the only diagnosis associated with increased resistance to COVID-19 was personal history of malignant neoplasm. One possible explanation for such finding is that patients with cancer history may be taking particular medications that in turn reduce their risk of

getting infected. Another possibility is that this group of patients is more careful in terms of social distancing practices and is less likely to be exposed to SARS-CoV-2. Although we attempted to mitigate the possibility of differential exposure by different groups of individuals by including in the analysis only the patients indicating exposure as the reason for testing, this possibility can never be eliminated due to reporting bias.

In addition, we observed higher number of comorbidities and medications per individual in the non-resistant cohort, when compared to resistant individuals, suggesting that resistant individuals may have better overall health.

The models we trained achieved low to moderate predictive performance, despite the application of a variety of feature selection and class balancing techniques. However, p-values obtained through Monte Carlo simulations are <0.001 for the testing set for all three models, indicating that the results are statistically significant and that, though somewhat weak, there are signals capable of distinguishing the resistant from non-resistant patients. We believe the XGBoost performed the best because it goes beyond logistic regression and random models forest models by correcting itself over time by increasing the weight of high-error points to improve its fit. This can sometimes lead to overfitting, but by testing it on two held-out testing sets, one of which was sampled from a population different from the one we trained the models on, we verify that it likely was not overfitting. However, it still wasn't a very reliable predictor despite all these efforts.

This work shows the feasibility of machine learning approaches to detect patients who are likely to be resistant to an emerging infection. A model with good discriminative ability may be used by public health professionals to enable better stratification of risk groups and improve surveillance. Similar tools have been developed previously for prediction of COVID-19 severity to improve care for patients hospitalized due to COVID-19. Given the high burden that was put onto the healthcare system early in COVID-19 pandemic, we believe that the resistance prediction tool may help allocate limited healthcare resources more efficiently in case of potential future outbreaks [29].

To our knowledge, there have been no studies published on predicting COVID-19 resistance using machine learning approaches, and our study demonstrated that this can be potentially achieved. In addition, this is the first study of COVID-19 resistance that includes negative cases in the analysis, and uses household information to infer COVID-19 exposure from the EHR.

## Limitations

It is important to recognize several limitations to our study. First, exposure to SARS-CoV-2 was self-reported through the COVID-19 testing questionnaire by the participant, and therefore, is a subject to reporting bias and is not completely reliable. The degree and duration of the exposure are not well-captured, and it is impossible to tell whether a subject was wearing personal protective equipment (e.g., mask) during the exposure, which could have protected the individual from infection. Second, there were very few participants in the high confidence of exposure cohort based on the HHI, which might have been due to stringent matching by longitude/latitude. In addition, those without a HHI match did not necessarily live alone or did not live with someone who tested positive; their cohabitors might just have never visited the Johns Hopkins Medical System and therefore, not be in the Johns Hopkins medical record. Moreover, only the COVID-19 tests done within the Johns Hopkins Medical System are included in the registry. Therefore, if a patient tested at a different facility, or self-tested with a Rapid Antigen test, the results are not captured in JH-CROWN, which may have led to underestimation of positive test results. Besides, some of the individuals included in our analysis

may have had unreported asymptomatic COVID-19 infection prior to their exposure, which may have provided temporary immunity to SARS-CoV-2 infection that was independent of the factors assessed in this study. Finally, the time frame of our study is limited, which means that some participants who were labeled as resistant may have tested positive for COVID-19 outside of this time frame. For this study, we labeled exposed individuals who consistently tested negative, but it is important to acknowledge resistant individuals within this time frame may not retain resistance over time given a larger period.

## Implications and future directions

In the future, we aim to evaluate how the model and feature associations identified using the JH-CROWN cohort can be generalized to the general population. More advanced association studies should be conducted to validate the risk factors identified to be associated with the resistance status. In addition, to further increase the confidence of exposure to SARS-CoV-2 of each participant, more detailed contact tracing information can potentially be leveraged. We hope that this model can be used in a clinical setting to predict the likelihood of an exposed patient of getting infected with SARS-CoV-2 and identify factors that affect that likelihood.

## Supporting information

**S1 Text.**
(ZIP)

## Acknowledgments

We wish to thank Dr. Dhananjay Vaidya, Dr. Jacky Jennings, Lisa Yanek, and Bahareh Modanloo from the Biostatistics, Epidemiology, and Data Management (BEAD) core group for their guidance and constructive comments on study design and analysis approaches. We also thank Kerry Smith and Michael Cook from the Center for Clinical Research Data Acquisition (CCDA) for their help in data extraction and data navigation. The data utilized were part of JH-CROWN: The COVID PMAP Registry, which is based on the contribution of many patients and clinicians.

## Author Contributions

**Conceptualization:** Kai-Wen K. Yang, Chloé F. Paris, Kevin T. Gorman, Ilia Rattsev, Rebecca H. Yoo, Yijia Chen, Casey Overby Taylor, Stuart C. Ray.

**Data curation:** Kai-Wen K. Yang, Chloé F. Paris, Kevin T. Gorman, Ilia Rattsev, Rebecca H. Yoo, Yijia Chen.

**Formal analysis:** Kai-Wen K. Yang, Chloé F. Paris, Kevin T. Gorman, Ilia Rattsev, Rebecca H. Yoo, Yijia Chen.

**Investigation:** Kai-Wen K. Yang, Chloé F. Paris, Kevin T. Gorman, Ilia Rattsev, Rebecca H. Yoo, Yijia Chen.

**Methodology:** Kai-Wen K. Yang, Chloé F. Paris, Kevin T. Gorman, Ilia Rattsev, Rebecca H. Yoo, Yijia Chen.

**Project administration:** Kai-Wen K. Yang.

**Supervision:** Jacob M. Desman, Tony Y. Wei, Joseph L. Greenstein, Casey Overby Taylor, Stuart C. Ray.

**Visualization:** Kai-Wen K. Yang, Chloé F. Paris, Kevin T. Gorman, Ilia Rattsev, Rebecca H. Yoo, Yijia Chen.

**Writing – original draft:** Kai-Wen K. Yang, Chloé F. Paris, Kevin T. Gorman, Ilia Rattsev, Rebecca H. Yoo, Yijia Chen.

**Writing – review & editing:** Kai-Wen K. Yang, Chloé F. Paris, Kevin T. Gorman, Ilia Rattsev, Rebecca H. Yoo, Yijia Chen, Jacob M. Desman, Tony Y. Wei, Joseph L. Greenstein, Casey Overby Taylor, Stuart C. Ray.

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
