## [Decision Letter · Decision Letter 0]

28 Sep 2022

PONE-D-22-23813Factors Associated with Resistance to SARS-CoV-2 Infection Discovered Using Large-scale Medical Record Data and Machine LearningPLOS ONE

Dear Dr. Ray,

Thank you for submitting your manuscript to PLOS ONE. After careful consideration, we feel that it has merit but does not fully meet PLOS ONE’s publication criteria as it currently stands. Therefore, we invite you to submit a revised version of the manuscript that addresses the points raised during the review process.

In my opinion adding Data Visualization as a part of the manuscript will help readers to understand the data better. Also reviewers have asked to include comparitive analysis with state of art algorithms.

We look forward to receiving your revised manuscript.

Kind regards,

Sathishkumar V E

Academic Editor

PLOS ONE

Journal Requirements:

"The data utilized were part of JH-CROWN: The COVID PMAP Registry, which is based on the contribution of many patients and clinicians and is funded by Hopkins inHealth, the Johns Hopkins Precision Medicine Program. Project-specific costs of data extraction were defrayed by funds from the Office of the Dean, JHU School of Medicine."

"The data utilized were part of JH-CROWN: The COVID PMAP Registry,

which is based on the contribution of many patients and clinicians and is funded by

Hopkins inHealth, the Johns Hopkins Precision Medicine Program. Project-specific

costs of data extraction were defrayed by funds from the Office of the Dean, JHU

School of Medicine."

"The data utilized were part of JH-CROWN: The COVID PMAP Registry, which is based on the contribution of many patients and clinicians and is funded by Hopkins in Health, the Johns Hopkins Precision Medicine Program. Project-specific costs of data extraction were defrayed by funds from the Office of the Dean, JHU School of Medicine."

6. Please include a caption for figure 3a and 3b.

Reviewers' comments:

Reviewer's Responses to Questions

**Comments to the Author**

1. Is the manuscript technically sound, and do the data support the conclusions?

Reviewer #1: No

Reviewer #2: Yes

Reviewer #3: Yes

2. Has the statistical analysis been performed appropriately and rigorously? 

Reviewer #1: No

Reviewer #2: Yes

Reviewer #3: Yes

3. Have the authors made all data underlying the findings in their manuscript fully available?

Reviewer #1: No

Reviewer #2: Yes

Reviewer #3: Yes

4. Is the manuscript presented in an intelligible fashion and written in standard English?

Reviewer #1: Yes

Reviewer #2: Yes

Reviewer #3: Yes

5. Review Comments to the Author

Reviewer #1: This project begins with a worthy question - why do some people get exposed to COVID, but not develop SARS-COV-2. The writing is good and the figures are fine.

Unfortunately, the data and analysis do not meet the needs of question. As the limitations section (appropriately) makes clear, there is simply an untenable amount of noise and bias in every variable in the dataset to answer this question. The exposure, outcome, and key predictors are all absolutely unreliable to address the underlying question they are hoping of them, given low testing rates, poorly-measured exposure, etc. ICD codes are rarely accurate, but especially given the crazy and unreliable primary care access that we all experienced in 2020. I just don't trust these data for these questions.

I was also concerned by the analysis, which had a lot of examples of what seemed to me to be opportunities for overfitting. These include running the same analysis with multiple different resampling and model-building techniques and multiple different tuning parameters, and using backward elimination without penalization. I am only too aware that this is common in machine learning, but it's a method of performing multiple comparisons without appropriate adjustment Similarly, "using five-fold validation to help tune the hyperparameters " has the same problem. Cross-validation is a validation technique, not a tuning technique.

Finally, I don't really understand the purpose the main goals of the analysis. For the classification model, what would we do with an effective classification tool? I'm not expecting this to be the end-all classification model, but it's nice to know why we're making a prediction.

Similarly, I don't really understand what we hoped to learn from the clustering. Were we hoping to find a clear biologic cause of resistance? To me, a cluster implies actual meaningful differences between groups, like Type 1 vs. Type 2 diabetes. Clustering algorithms like this are designed to find clusters in what are usually actually just putting lines around non-meaningful differences. I'm not convinced these clusters are meaningful and I'm not sure what to do with them if they are.

I do look forward to seeing further work in this field to understand why some patients did not develop SARS-COV-19.

Reviewer #2: The work seems to be very much appreciable. I have only a few points.

1. feature selection : why did not the authors mention the algorithm?

2. A few more recent references to be added

3. questionnaire to be placed in Appendix.

Reviewer #3: The overall presentation and results seems interesting.

What is the motivation of the proposed work? Research gaps, objectives of the proposed work should be clearly justified

Insert a figure demonstrating the overall steps involved.

This is a classification problem, so make a table summaizing the Accuracy, sensitivity, specificity, adn other performance metrics.

Authors used XGBoost model for model development. Authors are requested to compare XGboost algorithm performance with traditional algorithms and make a comparitive study. Authors are suggested to include more discussion on the results and also include some explanation regarding the justification to support why the proposed method is better in comparison towards other methods

Whether hyperparameter tuning performed? If yes what strategy is folllowed to select the best hyperparameters?

Results and discussion section should be improved.

Discuss about the correlation between the variables/features considered.

Explain why the current method was selected for the study, its importance and compare with traditional methods.

Does this kind of study have never attempted before? Justify this statement and give an appropriate explanation to do so in this paper.

Quality of figures is so important too. Please provide some high-resolution figures. Some figures have a poor resolution.

6. PLOS authors have the option to publish the peer review history of their article (what does this mean?). If published, this will include your full peer review and any attached files.

Reviewer #1: No

Reviewer #2: No

Reviewer #3: No

---

## [Author Response · Author response to Decision Letter 0]

31 Oct 2022

Please see detailed Response to Reviewers, which depends on formatting for clarity

Have added LaTex (.tex) file as "Other" type.

---

## [Decision Letter · Decision Letter 1]

17 Nov 2022

Factors Associated with Resistance to SARS-CoV-2 Infection Discovered Using Large-scale Medical Record Data and Machine Learning

PONE-D-22-23813R1

Dear Dr. Ray,

We’re pleased to inform you that your manuscript has been judged scientifically suitable for publication and will be formally accepted for publication once it meets all outstanding technical requirements.

Kind regards,

Sathishkumar V E

Academic Editor

PLOS ONE

Additional Editor Comments (optional):

Reviewers' comments:

Reviewer's Responses to Questions

**Comments to the Author**

1. If the authors have adequately addressed your comments raised in a previous round of review and you feel that this manuscript is now acceptable for publication, you may indicate that here to bypass the “Comments to the Author” section, enter your conflict of interest statement in the “Confidential to Editor” section, and submit your "Accept" recommendation.

Reviewer #3: (No Response)

2. Is the manuscript technically sound, and do the data support the conclusions?

Reviewer #3: (No Response)

3. Has the statistical analysis been performed appropriately and rigorously? 

Reviewer #3: (No Response)

4. Have the authors made all data underlying the findings in their manuscript fully available?

Reviewer #3: (No Response)

5. Is the manuscript presented in an intelligible fashion and written in standard English?

Reviewer #3: (No Response)

6. Review Comments to the Author

Reviewer #3: (No Response)

7. PLOS authors have the option to publish the peer review history of their article (what does this mean?). If published, this will include your full peer review and any attached files.

Reviewer #3: No

---

## [Editor Report · Acceptance letter]

28 Nov 2022

PONE-D-22-23813R1 

Factors associated with resistance to SARS-CoV-2 infection discovered using large-scale medical record data and machine learning 

Dear Dr. Ray:

I'm pleased to inform you that your manuscript has been deemed suitable for publication in PLOS ONE. Congratulations! Your manuscript is now with our production department. 

Kind regards, 

on behalf of

Dr. Sathishkumar V E 

Academic Editor

PLOS ONE